# Application of Initial Bias Estimation Method for Inertial Navigation System (INS)/Doppler Velocity Log (DVL) and INS/DVL/Gyrocompass Using Micro-Electro-Mechanical System Sensors

**DOI:** 10.3390/s22145334

**Published:** 2022-07-17

**Authors:** Gen Fukuda, Nobuaki Kubo

**Affiliations:** Department of Maritime Systems Engineering, Tokyo University of Marine Science and Technology, Tokyo 135-8533, Japan; nkubo@kaiyodai.ac.jp

**Keywords:** initial bias, trajectory generator, INS/DVL, INS/DVL/gyrocompass, MEMS

## Abstract

This article proposes a novel initial bias estimation method using a trajectory generator (TG). The accuracy of attitude and position estimation in navigation after using the inertial navigation system/Doppler velocity log (INS/DVL) and INS/DVL/gyrocompass (IDG) for 1 h were evaluated, and the results were compared to those obtained using the conventional Kalman filter (KF) estimation method. The probability of a horizontal position error < 1852 m (1 nautical mile) with a bias interval > 400 s was 100% and 9% for the TG and KF, respectively. In addition, the IDG average horizontal position errors over 1 h were 493 m and 507 m for the TG and KF, respectively. Moreover, the amount of variation was 2 m and 27 m for the TG and the KF, respectively. Thus, the proposed method is effective for initial bias estimation of INS/DVL and IDG using micro-electro-mechanical system sensors on a constantly moving vessel.

## 1. Introduction

The GNSS is the only instrument that can automatically estimate commercial ship positions. The GNSS is very cost effective and accurate; for example, real-time kinematic positioning (RTK)-GPS can estimate the position of a ship within centimeters. However, the vulnerability of GNSS signals to jamming and spoofing has been noted; recently, the Center for Advanced Defense Studies addressed the issue of spoofing in Russia and Syria [1]. Extensive studies have been conducted on developing effective jamming and spoofing detection systems [2,3,4]. However, currently, there are no affordable navigation systems that can be installed on commercial ships and that can automatically perform accurate position estimation after detecting an anomaly in the GNSS. To enhance the safety of autonomous ships, the position estimation system must be diversified.

Although a high-precision inertial navigation system (INS) is also available as an autonomous position estimation system, this system is expensive and can only be installed on a limited number of ships. Therefore, the use of low-cost inertial measurement units (IMUs) and Doppler velocity logs (DVLs) is being considered [5,6], as IMU accuracy has been improving in recent years, and DVLs are installed on many merchant ships. As the INS using a low-cost IMU causes drifts in the velocity estimate owing to noise, the combined use of IMUs and DVLs reduces this drift. In addition, as a DVL can only estimate velocity, it can be combined with an IMU to obtain the attitude. The authors of Ref. [7] stated that GNSS/IMU/DVL provides the maximum horizontal position error of less than 30 m, which is sufficiently large to make most maritime applications successful even in difficult jamming environments. Further, it has already been shown that the aforementioned accuracy limitation can be resolved for short-term operations. In a previous study, we combined an INS/DVL system using inexpensive inertial sensors and obtained 1 h of verified operation results on a ship [8]. In that study, the initial bias was estimated from the sensor values using the Kalman filter (KF) of the INS/GPS. The KF is an optimal filter based on a state–space model, and if the model is provided, a theoretically guaranteed optimal filter can be obtained through a systematic procedure. However, owing to the high dependence of the model on environmental factors, and particularly on the temperature, frequent calibration may be required for an accurate bias estimation using low-cost micro-electro-mechanical system (MEMS) sensors [9].

In INS/DVL, when GPS position correction is unavailable, the initial bias of the angular rate and acceleration sensors significantly affect the position estimation accuracy. In a previous study [8], initial bias estimation was performed using angular velocity and acceleration estimates from the KF, but adjusting the parameters of the KF was time consuming. Furthermore, using this adjustment to perform an estimate with similar accuracy on the next voyage was expected to be difficult for the reasons discussed above. In particular, it is difficult to compensate with high precision for the initial bias that occurs when the power is turned on. This issue can be solved by frequently calibrating the sensors installed on the ship, but this solution is unrealistic. Therefore, we decided to investigate a more reproducible method for initial bias estimation, which has a large impact on position estimation error in INS/DVL that does not directly use values from the KF. The advantages and disadvantages of INS/GPS with MEMS IMU can be summarized as follows: (a) GPS corrections can be used to obtain highly accurate position estimates; (b) the speed can be estimated without drift errors using GPS speed corrections; (c) the necessary accuracy for roll and pitch can be estimated using only a MEMS IMU [8]; and (d) improving the accuracy of heading estimation using only an IMU on a vessel with drift is difficult. Considering the advantages (a–c), the angular velocity and acceleration estimates from the trajectory generator (TG) [10] for simulating inertial navigation systems could be used for initial bias estimation. Using the TG, the parameters necessary for inertial navigation calculations, including noise-free angular velocity and acceleration, can be obtained by inputting the desired position, attitude, and velocity. The disadvantage (d) can be compensated using a gyrocompass onboard the vessel to include heading correction. This report proposes an initial bias estimation method that can be used for vessels with motion in roll and pitch using angular velocity and acceleration estimated from TG and comparing them with measurements obtained from an IMU. For the angular rate sensor bias, an initial bias estimation has been previously performed using an inverse algorithm for attitude update [11], which utilizes the fact that attitude drift does not occur even for low-cost IMUs. In this study, bias estimation was also performed for the accelerometer by obtaining an acceleration estimate that considers the posture. For acceleration, the accelerometer bias estimation using zero velocity update has been performed for underwater vehicles [12], but it is difficult to perform such estimation for constantly moving ships.

As the development of autonomous vessels continues, no solution has been proposed for the case of GNSS failure. In ships, the redundancy of the position estimation system is an issue that must be solved as soon as possible. Some previous studies, such as ours, have attempted to solve this problem by combining low-cost IMUs and DVLs and comparing, for example, KFs [5,13]. However, according to our research, no previous article has proposed an initial bias estimation for the use of low-cost IMUs on ships or presented an evaluation of its impact on position estimation errors in a non-GNSS environment. What we want to show in this paper is that the initial bias estimation presented in this paper can be used to achieve unprecedented accuracy in non-GNSS INS/DVL using low-cost IMUs, where the accuracy of position estimation is greatly affected by the initial bias. For vessels that rely solely on GNSS for automatic position estimation, there is an urgent need to make automatic position estimation systems redundant for safety reasons, and we believe that this information will be important for other researchers working on this issue. Thus, we hope that this study will contribute to redundant position estimation systems for ships.

The rest of this paper is organized as follows. Section 2 provides an outline of the system. Section 3 describes the experimental method used in an actual scenario. From the actual measurements, the initial bias was estimated using the proposed and conventional KF methods. The obtained estimates of the initial bias were used as inputs into the INS/DVL and INS/DVL/gyrocompass (IDG), respectively, and the calculated horizontal error after 1 h was obtained. Section 4 and Section 5 present the discussion and conclusion, respectively.

## 2. System Using Initial Bias Estimation by Inversion of Inertial Navigation Calculations

A TG is a typical example of using inverse inertial navigation calculations. It is a software simulator that is employed to generate numerical data of navigation parameters corresponding to a specified trajectory profile [10]. The generated data are utilized in various simulation programs to verify the INS-related software operations. Basic outputs include attitude, velocity, and position navigation data, as well as acceleration and angular rate sensor estimates [10]. In addition, the inertial explorer [14], which estimates various sensor outputs from actual measurement data, is available, although the number of sensors that can be used is limited. However, none of these systems are designed for real-time operation.

Figure 1 shows the initial bias estimation scheme used in the experimental results in Section 3. The TG estimates the bias for a 30 min period extending up to 10 min before the start of INS/DVL and IDG. For the bias estimation, the time at switching to INS/DVL was set to 0 s, and 1799 biases were estimated up to 1800 s by changing the data interval every second; these were used for INS/DVL and IDG. The time and data intervals for bias estimation in the KF are the same as those in the TG. The “Est.” stands for “Estimation”.

This section describes how the velocity, attitude, and position estimated by the INS/GPS/gyrocompass (IGG) are input into the TG-based system [15] and how the estimated acceleration and angular velocity are used to estimate the initial bias and applied to INS/DVL and IDG. The INS/DVL is the same as in the previous study [8], and therefore, its derivation process is omitted.

The earth fixed coordinate frame (E); navigation coordinate frame of the azimuth wander type (N); locally level coordinate frame (L); locally level geographic coordinate frame (Geo); north, east, and down (NED); and body frame (B) were used according to the coordinate system in Ref. [16]. Table 1 lists the abbreviations used in this paper.

### 2.1. Initial Bias Estimation

The initial bias estimation between the proposed TG and the KF when switching from IGG to INS/DVL can be calculated as follows. As the TG is used to estimate the IGG attitude, the initial bias is calculated from the average of the difference between the TG and the sensor value corresponding to that time. For the accelerometers, the difference between the average values of the TG and the sensor is used because the correction is applied. In this study, $t_{\mathrm{End}}$ = 1800 s.
(1)$${\underline{\omega }}_{{\mathrm{Bias}}_{\mathrm{TG}}}=\frac{\sum_{i=n}^{End}{\underline{\omega }}_{{\mathrm{IB}}_{S\left(t_{i}\right)}}^{B}-\sum_{i=n}^{End}{\underline{\omega }}_{{\mathrm{IB}}_{\mathrm{TG}\left(t_{i}\right)}}^{B}}{t_{\mathrm{End}}-t_{n}}$$
(2)$${\underline{\omega }}_{{\mathrm{Bias}}_{\mathrm{KF}}}=\frac{\sum_{i=n}^{End}{\underline{\omega }}_{{\mathrm{IB}}_{S\left(t_{i}\right)}}^{B}-\sum_{i=n}^{End}{\underline{\omega }}_{{\mathrm{IB}}_{\mathrm{KF}\left(t_{i}\right)}}^{B}}{t_{\mathrm{End}}-t_{n}}$$

Similarly, for accelerometers
(3)$${\underline{a}}_{{\mathrm{Bias}}_{\mathrm{TG}}}=\frac{\sum_{i=n}^{End}{\underline{a}}_{{\mathrm{SF}}_{S\left(t_{i}\right)}}^{B}-\sum_{i=n}^{End}\;{\underline{a}}_{{\mathrm{SF}}_{\mathrm{TG}\left(t_{i}\right)}}^{B}}{t_{\mathrm{End}}-t_{n}}$$
(4)$${\underline{a}}_{{\mathrm{Bias}}_{\mathrm{KF}}}=\frac{\sum_{i=n}^{End}\;{\underline{a}}_{{\mathrm{SF}}_{S\left(t_{i}\right)}}^{B}-\sum_{i=n}^{End}\;{\underline{a}}_{{\mathrm{SF}}_{\mathrm{KF}\left(t_{i}\right)}}^{B}}{t_{\mathrm{End}}-t_{n}}.$$

${\underline{\omega }}_{\mathrm{Bias}}$: Estimated initial bias of the angular rate sensor.

${\underline{a}}_{\mathrm{Bias}}$: Initial bias estimate of the acceleration sensor.

${\underline{\omega }}_{{\mathrm{IB}}_{\left(t_{i}\right)}}^{B}$: Angular rate vector at time $t_{i}$ of frame B relative to the inertial space expressed in the B frame axes.

${\underline{a}}_{{\mathrm{SF}}_{\left(t_{i}\right)}}^{B}$: Specific force acceleration vector at time $t_{i}$ in frame B.

S: Angular rate sensor or acceleration sensor measurements.

TG: Angular rate or acceleration estimated by the TG.

KF: Angular rate or acceleration estimated by the KF.

$t_{\mathrm{End}}$: End time of estimation by the TG.

$T_{n}$: Start time of the data used for initial bias estimation (i.e., in this study, $t_{n}$ ranged from 1 to 1799).

### 2.2. IGG and IDG Integration

To use the TG regeneration process [10] proposed in this study, it is necessary to input the attitude, initial velocity, and target position data into each segment. Therefore, IGG estimation data were used in this study. The basic configuration of the INS/GPS system is based on NaveGo [17,18], which uses a 100 Hz IMU input and a 5 Hz GPS signal. As in a previous report [8], the inertial navigation calculation part of NaveGo was changed from Equation (17.2.3.1-2) to Equation (17.2.3.1-28) in Ref. [10]. Because NaveGo uses the NED coordinate system in the KF calculation, it was converted into the NED coordinate system using the Euler and Wander angles calculated in the L-coordinate system equation.

The error equation for the direction cosine matrix relating coordinate frames B and L($C_{B}^{L}$) can be derived in the navigation coordinate frame and local-level coordinate frame (wander-azimuth), as shown in Equations (5) and (6) [16,19].
(5)$$\dot{{\underline{\Gamma }}^{L}}=-\left({\underline{\omega }}_{\mathrm{IL}}^{L}\times \right)\;{\underline{\gamma }}^{L}-C_{B}^{L}\delta \;{\overset{˜}{\underset{\_}{\omega }}}_{IB}^{B}-\;{\underline{e}}^{L}\times {\underline{\omega }}_{\mathrm{IE}}^{L}+C_{N}^{L}\frac{1}{R_{E}}\left({\underline{u}}_{\mathrm{ZN}}^{N}\times {\delta \;\underline{v}}^{N}\right)$$
(6)$${\delta C}_{B}^{L}=-{\;\underline{\gamma }}^{L}\times C_{B}^{L}$$
(7)$$\delta {\overset{˜}{\underset{\_}{\omega }}}_{IB}^{B}=\eta_{g}+b_{g}+\eta_{g\delta b}$$
where $\eta_{g}$, $b_{g}$, and $\eta_{g\delta b}$ are an angle random walk noise, static bias, and discrete sequence related to bias instability ${\delta \underline{b}}_{g}$, respectively [20]; ${\underline{\gamma }}^{L}$ is the small-angle rotation vector error associated with $\delta C_{B}^{L}$; $C_{B}^{L}$ is the direction cosine matrix that transforms a vector from its B frame projection form into its L frame projection form; $\delta {\overset{˜}{\underset{\_}{\omega }}}_{IB}^{B}$ is the gyroscope sensor measurement errors; ${\underline{\omega }}_{\mathrm{IL}}^{L}$ and ${\underline{\omega }}_{\mathrm{IE}}^{N}$, are the angular rates of the local-level coordinate frame relative to the inertial frame, and the earth frame relative to the inertial frame, respectively; ${\underline{u}}_{\mathrm{ZN}}^{N}$ is the unit vector relative to the earth in N frame axes [16].

The error equation for the $\delta {\overset{˙}{\underset{\_}{v}}}^{L}$ is derived taking into account ${\delta \underline{v}}^{L}$, which is the error in velocity relative to the earth measured in the N Frame part:(8)$$\delta {\overset{˙}{\underset{\_}{v}}}^{L}=\left(C_{B}^{L}{\underline{a}}_{\mathrm{SF}}^{B}\right)\times {\underline{\gamma }}^{L}+C_{B}^{L}\delta {\overset{˜}{\underset{\_}{a}}}_{\mathrm{SF}}^{B}-C_{N}^{L}\left(2{\underline{\omega }}_{\mathrm{IE}}^{N}+{\underline{\omega }}_{\mathrm{EN}}^{N}\right)\times {\delta \underline{v}}^{N}+C_{N}^{L}\left({\underline{v}}^{N}\times 2{\underline{\omega }}_{\mathrm{IE}}^{N}\right)\times {\underline{e}}^{L}$$
(9)$$\delta {\overset{˜}{\underset{\_}{a}}}_{\mathrm{SF}}^{B}=\eta_{f}+b_{f}+\eta_{f\delta b}$$
where, $\eta_{f}$, $b_{f}$, and $\eta_{f\delta b}$ are a velocity random walk noise, static bias, and discrete sequence related to bias instability ${\delta \underline{b}}_{f}$, respectively [20]; $C_{N}^{L}$ is the direction cosine matrix that transforms vectors from N to L frame [16]; ${\underline{a}}_{\mathrm{SF}}^{B}$ is the specific force acceleration vector in frame B [16]; $\delta {\overset{˜}{\underset{\_}{a}}}_{\mathrm{SF}}^{B}$ is the acceleration sensor measurement errors. ${\underline{\omega }}_{\mathrm{EN}}^{N}$ is the navigation frame relative to the earth frame.

${\underline{e}}^{L}$, which is the small angle rotation vector associated with ${\delta C}_{E}^{L}$ [19], is derived using $R_{E}$, the radial distance from the center of the earth to the INS:(10)$$\dot{{\underline{e}}^{L}}={\underline{e}}^{L}\times {\;\underline{\omega }}_{\mathrm{EL}}^{L}+C_{N}^{L}\frac{1}{R_{E}}\left({\underline{u}}_{\mathrm{ZN}}^{N}\times {\delta \underline{v}}^{N}\right)$$
(11)$${\delta C}_{E}^{L}=-{\underline{e}}^{L}\times C_{E}^{L}$$

The INS error state vectors $\delta {\hat{\underline{x}}}_{\mathrm{IGG}}$ for IGG and INS error state vectors $\delta {\hat{\underline{x}}}_{\mathrm{IDG}}$ for IDG can be summarized as follows:(12)$$\delta {\hat{\underline{x}}}_{\mathrm{IGG}}={\left[{\underline{\gamma }}^{L},{\delta \underline{v}}^{L},{\underline{e}}^{L},{\delta \underline{b}}_{g},{\delta \underline{b}}_{f}\right]}^{T}$$
(13)$$\delta {\hat{\underline{x}}}_{\mathrm{IDG}}={\left[{\underline{\gamma }}^{L},{\delta \underline{v}}^{L},{\delta \underline{b}}_{g},{\delta \underline{b}}_{f}\right]}^{T}.$$

$\delta {\underline{b}}_{g}$ and ${\delta \underline{b}}_{f}$: Bias instability estimation vectors for the gyroscope and the accelerometer, respectively. The continuous and discrete state–space models of the system can be used as follows [21,22]:(14)$$\delta {\dot{\hat{x}}}_{\left(t\right)}=F_{\left(t\right)}{\delta \hat{x}}_{\left(t\right)}+G_{\left(t\right)}u_{\left(t\right)}+\zeta_{\left(t\right)}$$
(15)$$\delta {\hat{y}}_{\left(t\right)}=H\delta {\hat{x}}_{\left(t\right)}+\nu_{\left(t\right)}$$
(16)$${\delta \hat{x}}_{\left(+\right)}=\Phi \delta \hat{x}+\mathrm{Gu}+\zeta$$
(17)$$\delta \hat{y}=H\delta \hat{x}+\nu .$$

Vectors ζ and v are known as the driving noise and measurement noise with zero-mean Gaussian white noise, respectively [21]. The vector u and the covariance matrix, Q, were set up according to a previous study [8]. The state–space matrices for IGG and IDG are
(18)$$F_{\left(t\right)\left\{15\times 15\right\},\mathrm{IGG}}=\left[\begin{array}{ccccc} F1 & F2 & F3 & -{\hat{C}}_{B}^{L} & 0_{3}\\ F4 & F5 & F6 & 0_{3} & {\hat{C}}_{B}^{L}\\ 0_{3} & F7 & F8 & 0_{3} & 0_{3}\\ 0_{3} & 0_{3} & 0_{3} & F9 & 0_{3}\\ 0_{3} & 0_{3} & 0_{3} & 0_{3} & F10 \end{array}\right]$$
(19)$$F_{\left(t\right)\left\{12\times 12\right\},\mathrm{IDG}}=\left[\begin{array}{cccc} F1 & F2 & -{\hat{C}}_{B}^{L} & 0_{3}\\ F4 & F5 & 0_{3} & {\hat{C}}_{B}^{L}\\ 0_{3} & 0_{3} & F9 & 0_{3}\\ 0_{3} & 0_{3} & 0_{3} & F10 \end{array}\right]$$
(20)$$F1=-\left({\underline{\omega }}_{\mathrm{IL}}^{L}\times \right)$$
(21)$$F2=\frac{1}{R}C_{N}^{L}\left({\underline{u}}_{\mathrm{ZN}}^{N}\times \right)$$
(22)$$F3=\left({\;\underline{\omega }}_{\mathrm{IE}}^{L}\times \right)$$
(23)$$F4={\hat{C}}_{B}^{L}{\underline{a}}_{\mathrm{SF}}^{B}\times$$
(24)$$F5=-C_{N}^{L}\left(2{\;\underline{\omega }}_{\mathrm{IE}}^{N}+{\underline{\omega }}_{\mathrm{EN}}^{N}\right)\times$$
(25)$$F6=C_{N}^{L}\left({\underline{v}}^{N}\times 2{\;\underline{\omega }}_{\mathrm{IE}}^{N}\right)\times$$
(26)$$F7=\frac{1}{R}C_{N}^{L}\left({\underline{u}}_{\mathrm{ZN}}^{N}\times \right)$$
(27)$$F8=-{\underline{\omega }}_{\mathrm{EN}}^{N}\times$$
(28)$$F9=-\frac{1}{\tau_{g}}$$
(29)$$F10=-\frac{1}{\tau_{f}}$$
(30)$${\hat{C}}_{B}^{L}=\left(I_{3}-\Gamma^{L}\right)C_{B}^{L},$$
where $0_{3}$ is a 3 × 3 zero matrix; ${\underline{u}}_{\mathrm{ZN}}^{N}$ is a unit vector relative to the earth in the N frame axes [16]; $\tau_{g}$ and $\tau_{f}$ are the correlation times of the dynamic accelerometer and gyroscope biases, respectively [23]; ${\hat{C}}_{B}^{L}$ is the direction cosine matrix with an error; and $\Gamma^{L}$ is a skew-symmetric operator associated with ${\underline{\gamma }}^{L}$.
(31)$$R_{\left\{9\times 9\right\},\mathrm{IGG}}=\left[\begin{array}{ccc} \mathrm{diag}({\mathrm{std}}_{\mathrm{Gyro}}{}^{2}) & 0_{3} & 0_{3}\\ 0_{3} & \mathrm{diag}({\mathrm{std}}_{\mathrm{GPSv}}{}^{2}) & 0_{3}\\ 0_{3} & 0_{3} & \mathrm{diag}({\mathrm{std}}_{\mathrm{GPSm}}{}^{2}) \end{array}\right]$$
(32)$$R_{\left\{6\times 6\right\},\mathrm{IDG}}=\left[\begin{array}{cc} \mathrm{diag}({\mathrm{std}}_{\mathrm{Gyro}}{}^{2}) & 0_{3}\\ 0_{3} & \mathrm{diag}\left({\mathrm{std}}_{\mathrm{DVL}}{}^{2}\right) \end{array}\right]$$
(33)$${\mathrm{std}}_{\mathrm{Gyro}}=\left[\begin{array}{ccc} 1 & 1 & \sigma_{\mathrm{GC}} \end{array}\right]$$
(34)$${\mathrm{std}}_{\mathrm{GPSv}}=\left[\begin{array}{ccc} \sigma_{v_{N}} & \sigma_{v_{E}} & \sigma_{v_{D}} \end{array}\right]$$
(35)$${\mathrm{std}}_{\mathrm{GPSm}}=\left[\begin{array}{ccc} \sigma_{\mathrm{lat}} & \sigma_{\mathrm{Lon}} & \sigma_{\mathrm{Alt}} \end{array}\right]$$
(36)$${\mathrm{std}}_{\mathrm{DVL}}=\left[\begin{array}{ccc} \sigma_{V_{x}}{}^{2} & \sigma_{V_{y}}{}^{2} & 1 \end{array}\right]$$
where ${\mathrm{std}}_{\mathrm{Gyro}}$, ${\mathrm{std}}_{\mathrm{GPSv}}$, ${\mathrm{std}}_{\mathrm{GPSm}}$, and ${\mathrm{std}}_{\mathrm{DVL}}$ are the standard deviations of the gyrocompass, NED-coordinate GPS velocity, GPS position, and DVL velocity, respectively.

The IGG measurement models of the KF for the heading, velocity, and position are as follows:(37)$${\delta \hat{y}}_{\mathrm{IGG}}=\left[\begin{array}{ccc} {\delta \hat{y}}_{H}^{T} & {\delta \hat{y}}_{v}^{T} & {\delta \hat{y}}_{p}^{T} \end{array}\right]$$
(38)$${\delta \hat{y}}_{\mathrm{IDG}}=\left[\begin{array}{cc} {\delta \hat{y}}_{H}^{T} & {\delta \hat{y}}_{\mathrm{INS}/\mathrm{DVL}} \end{array}\right]$$
(39)$${\delta \hat{y}}_{\mathrm{INS}/\mathrm{DVL}}={\hat{\underline{v}}}_{\mathrm{INS}}^{L}-{\hat{\underline{v}}}_{\mathrm{DVL}}^{L}={\delta \underline{v}}_{\mathrm{INS}}^{L}-\left[\left(C_{B}^{L}{\left(C_{B}^{D}\right)}^{T}{\underline{v}}_{\mathrm{DVL}}^{D}\right)\times \right]{\underline{\gamma }}^{L}$$
(40)$${\delta \hat{y}}_{H}=\left[0\;0{\;\mathrm{Hdif}}_{t}\right]$$
(41)$${\mathrm{Hdif}}_{t}={\mathrm{Gyrocompass}}_{\left(t\right)}-{\mathrm{yaw}}_{\left(t\right)}$$
(42)$${\delta \hat{y}}_{v}={\hat{\underline{v}}}_{\mathrm{INS}}^{\mathrm{NED}}-{\hat{\underline{v}}}_{\mathrm{GPS}}^{\mathrm{NED}}$$
(43)$${\delta \hat{y}}_{p}={\hat{T}}_{p}^{r}\left({\hat{\underline{p}}}_{\mathrm{INS}}^{L}-{\hat{\underline{p}}}_{\mathrm{GPS}}^{L}\right)+{\hat{C}}_{B}^{L}l_{arm}.$$

${\mathrm{Hdif}}_{t}$ is heading difference between the gyrocompass and yaw calculated by INS at time t. Please note that the equation 39 was taken from reference [24].

The gyrocompass is assumed to be aligned with frame B (in the case of this study, the right-hand system with the bow direction was considered as the *x*-axis). Thus, the measurement matrix, H, was calculated as
(44)$$H_{\left(t\right)\left\{9\times 15\right\},\mathrm{IGG}}=\left[\begin{array}{c} I_{3}\\ 0_{3}\\ 0_{3} \end{array}\begin{array}{c} 0_{3}\\ I_{3}\\ 0_{3} \end{array}\begin{array}{c} 0_{3}\\ 0_{3}\\ {\hat{T}}_{p}^{r} \end{array}\begin{array}{c} 0_{3}\\ 0_{3}\\ 0_{3} \end{array}\begin{array}{c} 0_{3}\\ 0_{3}\\ 0_{3} \end{array}\right]$$
(45)$$H_{\left(t\right)\left\{6\times 15\right\},\mathrm{IDG}}=\left[\begin{array}{c} I_{3}\\ -C_{B_{\left(t\right)}}^{L}{\left(C_{B}^{D}\right)}^{T}{\underline{V}}_{{\mathrm{DVL}}_{\left(t\right)}}^{D} \end{array}\begin{array}{c} 0_{3}\\ I_{3} \end{array}\begin{array}{c} 0_{3}\\ 0_{3} \end{array}\begin{array}{c} 0_{3}\\ 0_{3} \end{array}\begin{array}{c} 0_{3}\\ 0_{3} \end{array}\right]$$

$I_{3}$: 3 × 3 identity matrix.

${\hat{T}}_{p}^{r}$: Curvilinear-to-Cartesian transformation matrix [21].

$C_{B}^{D}$: Direction cosine matrix used as the misalignment matrix between frame B and DVL.

${\underline{V}}_{\mathrm{DVL}}^{D}$: Doppler velocity log.

$C_{B}^{L}$: Direction cosine matrix relating coordinate frames B and L.

### 2.3. Acceleration and Angular Velocity Estimation Using Inverted Form of INS Calculation

The velocity, attitude, and position estimated by (IGG) as described in Section 2.3 are input into the TG-based system [15] to obtain the estimated acceleration and angular velocity. The details are described in the authors’ previous study [15], but since the paper is in Japanese, this section briefly describes the acceleration and angular velocity estimation process.

In one segment, while adjusting the velocity, the data set (angular velocity and acceleration) is determined when the IGG and TG positions are the closest to each other, as shown in Figure 2. The part indicated as TG in the figure performs the calculation process shown in Equations (17.2.3.1-29), (17.2.3.1-30), and (17.2.3.1-31) of Ref. [10]. $C_{B_{\left(t\right)}}^{L}$ is the attitude matrix of the TG at time t, calculated by simultaneously using the attitudes (ϕ (roll), θ (pitch), and ψ (yaw)) in the IGG L-coordinate system [16]. The calculated $C_{B_{\left(t\right)}}^{L}$ is substituted into ${\left(C_{B}^{L}\right)}_{m}$ in Equation (17.2.3.1-29) from Ref. [10]. Similarly, the user specifies a velocity, ${\underline{v}}_{\left(t\right)}^{N}$
. As the target position is specified by the latitude and longitude, the system adjusts ${\underline{v}}_{\left(t\right)}^{\mathrm{Geo}}$ from the velocity, ${\underline{v}}_{\left(t\right)}^{\mathrm{NED}}$, in the NED coordinate system at time t, calculated by the IGG, which is performed by adding or subtracting the amount of adjustment required to reach the desired position to ${\underline{v}}_{\mathrm{IGG}\left(t\right)}^{\mathrm{NED}}$ at time t and converting it into ${\underline{v}}_{\left(t\right)}^{\mathrm{Geo}}$ according to the definition in Ref. [16]. The calculated ${\underline{v}}_{\left(t\right)}^{N}$ is substituted into ${\underline{v}}_{m}^{N}$ in Equation (17.2.3.1-30) from Ref. [10]. Abbreviations “Angul. velo.” and “Acc.” stand for “Angular velocity” and “Acceleration”, respectively.

## 3. Experiment Outline and Results

This section provides an overview of the experimental observations and estimation results of the initial bias obtained using the proposed method. It also presents the estimation results obtained by INS/DVL and IDG when the proposed method and KF were used to determine the initial bias, respectively.

### 3.1. Initial Bias Estimation

To validate the algorithm described in Section 2, the following experiment was conducted. Here, it was assumed that the interference was detected while the ship was navigating in Tokyo Bay, and the speed and attitude of the IGG before the interference were used as the initial values for INS/DVL and IDG. Figure 3 illustrates the track of this experiment. The IGG segment is shown in white, the TG estimation segment (30 min) is shown in green, and the INS/DVL and IDG estimation segment (60 min) is shown in blue.

The sensors used in this experiment were the same as those employed in a previous study [8]. The information about the sensors utilized in this experiment is summarized in Table 2 and Table 3. The IMU, namely, “CSG-MG100” [25], manufactured by Tokyo Aircraft Instrument Co., Ltd. (Tokyo, Japan) was used for time synchronization with the GNSS. This IMU can detect the acceleration and angular velocity along three axes. In the experiment, the IMU was set up with the bow direction as the *x*-axis, the starboard direction perpendicular to the bow direction as the *y*-axis, and the vertical downward direction as the *z*-axis. A four-beam DVL ATLAS DOLOG SYSTEM was installed at the bottom of the ship in front of the bow thruster. The DVL can detect the ground velocity in the forward/backward and left/right directions up to a depth of 200 m. The reference for the estimated position was the result of RTK positioning between the Trimble Net R9 Marine Network Reference Station and Trimble SPS855 receiver onboard the vessel. The roll and pitch reference values were sampled at 1 Hz using a JCS7402-A [26]. “Freq.” stands for “Frequency” in Table 2.

It should be noted that we could not obtain a reference value for the bow heading because we did not maintain a more accurate measuring instrument than the TG-5000 [27] shown in Table 3 as a reference. However, this situation does not affect the accuracy of the proposed initial bias estimation method, which is the basis of the discussion in this paper. In a previous experiment [8], the accuracy was verified by deliberately complicating the motion of the ship. Because the correction of the *z*-axis angular velocity bias due to the 360° turn of the ship was confirmed, in this experiment, the route was chosen such that the bias component would not be lost because of the 360° turn of the ship. In the cases in which the true value is unknown or the reference value is expected to contain errors, we use the term “difference” instead of “error”. 

### 3.2. Estimation Results from Inverse Inertial Navigation Calculations

It is assumed that at each time the necessary estimates were obtained from the IGG, the TG was processed in parallel with a slight delay relative to the IGG. Therefore, an initial bias estimation was performed using the TG estimates from 30 min to 10 min before the start of INS/DVL and IDG. The acceleration and angular velocity were estimated at 1 Hz for 30 min. Although it varies depending on the system requirements and the environment, for reference, it was determined that 30 min of estimation by the TG took 481.39 s. In addition, because the estimation time increased significantly, no attitude adjustment was performed.

#### 3.2.1. Comparison between IGG and Reference Estimations

The position and attitude of the IGG used for the TG estimation were compared with those of the RTK-GPS and fiber optic gyroscope (FOG) as references. As shown in Figure 4, the maximum horizontal error and average error of the IGG were 1.86 m and 0.4 m, respectively. These errors were within the error range shown in Table 1.

Figure 5a presents the rolls of IGG and FOG, and Figure 5b provides the root-mean-square (RMS) differences between the rolls of IGG and FOG. Figure 5c shows the pitches of IGG and FOG, and Figure 5d depicts the RMS values of the difference between the pitches of IGG and FOG. The maximum differences in the roll and pitch were measured to be 1.79° and 0.67°, respectively; moreover, the average differences were 1.09° and 0.24°, respectively. The FOG was fixed near the bottom of the ship, and the IMU was set up on a desk in the laboratory under the bridge. This difference between the installation conditions resulted in discrepancies, particularly in the roll values. The attitude could be used for the TG because of the absence of a drift component.

#### 3.2.2. Comparison of the TG and IGG Estimates

The difference in the horizontal positions estimated by the INS using the acceleration and angular velocity determined by the TG and the position of the IGG input as the target are shown in Figure 6. The maximum and average differences were measured to be 0.15 and 0.04 m, respectively.

#### 3.2.3. Estimates of Angular Velocity and Acceleration (Specific Force)

The actual angular velocities measured by the IMU, the values estimated by the TG, and their corresponding differences along each axis are shown in Figure 7. The maximum differences along the x, y, and z axes are 1.15, 0.98, and 0.46°/s, and the average RMS values are 0.19, 0.25, and 0.13°/s, respectively.

The actual acceleration measured by the IMU, acceleration estimated by the TG, and difference between them along each axis are shown in Figure 8. The maximum differences along the x, y, and z axes are 0.68, 1.08, and 0.32 m/s^2^, respectively, and the average differences are 0.47, 0.70, and 0.13 m/s^2^, respectively.

### 3.3. Comparison of Estimates Based on the Interval of Data Used to Estimate the Initial Bias

The initial bias estimation was conducted according to the scheme presented in Figure 1 in Section 2. Figure 9 and Figure 10 show the initial bias estimates and data interval relationships obtained by the TG and KF for angular velocity and acceleration, respectively. In the figures, the bias estimation by TG is shown in (a)–(c) and that by KF is shown in (d)–(f). For the angular velocity, when the data interval used for bias estimation is more than 400 s, the respective averages for the x, y, and z axes are −1.66 × 10^−2^, 2.30 × 10^−1^, and 1.23 × 10^−1^°/s for the TG, and −1.75 × 10^−2^, 2.37 × 10^−1^, and 1.59 × 10^−1^°/s for the KF. For acceleration, when the data interval used for bias estimation is more than 400 s, the averages for the x, y, and z axes are 4.74 × 10^−1^, 7.09 × 10^−1^, and −1.32 × 10^−1^ m/s^2^ for the TG, and 4.62 × 10^−1^, 8.23 × 10^−1^, and −1.23 × 10^−1^ m/s^2^ for the KF.

#### 3.3.1. Comparison of Results with Initial Bias Estimation Using the TG and KF

Figure 11 presents the RMS difference between the roll and pitch estimated by the INS/DVL and FOG during a 1 h voyage when the initial bias estimates from the TG and KF were used. The bias estimation was performed in 1 s intervals; however, in Figure 11, markers are placed every 1 min for easier viewing. The maximum and minimum differences in the RMS roll and pitch for the TG and KF are summarized in Table 4. When the data interval used for bias estimation is greater than 400 s, the average roll and pitch are 1.13° and 0.24° for TG and 1.78° and 0.18° for KF, respectively. For bias estimation by the TG, the difference converges when the bias estimation time for the roll is more than 400 s, but this trend is not observed for the KF. For pitch, the difference between the results obtained using TG and KF is 5.73 × 10^−2^°, and based on the accuracy of FOG in Table 1, we concluded that the estimation results were the same in this case.

Figure 12 shows the RMS difference between the roll and pitch of IDG and FOG for a 1 h voyage when using the initial bias estimates from the TG and KF. The interval parameters are the same as those specified in the previous paragraph. The maximum and minimum differences in RMS roll and pitch for the TG and KF are summarized in Table 5. When the data length used for bias estimation is greater than 400 s, the roll difference between the maximum and minimum values is 0.02°, which is almost the same when TG is used. In contrast, a variation of 0.13° is observed when KF is used. For pitch, the RMS mean difference between TG and KF is 0.06°. Based on the accuracy of the FOG used for comparison, it was determined that there was no estimated difference between TG and KF. “Ave.” in Table 5 and throughout the paper stands for “Average”.

#### 3.3.2. Initial Bias Estimation and Position Estimation Results Using TG and KF

The errors in the position (*y*-axis) estimated by INS/DVL and IDG when the RTK-GPS estimated position was used as the reference value and the TG and KF after 1 h shown in Figure 13 for each data length (*x*-axis) were utilized for bias estimation. The maximum and minimum differences in the horizontal positions for the INS/DVL for the TG and KF are summarized in Table 6. For INS/DVL, the horizontal position error in 1 h is less than 1852 m (1 nautical mile) with a probability of 100% for the TG and 6% for the KF. For IDG, the amount of variation is 2.4 m for the TG compared to 26.7 m for the KF. “Err” stands for “Error” in Figure 13.

Figure 14 shows the relationship between the *z*-axis angular velocity bias with a data length of 400 s or longer by TG and the horizontal error after 1 h by INS/DVL and IDG. For INS/DVL, the minimum error is 340 m/h, as shown in Figure 14a, and for IDG, the minimum error is 492 m/h and the average error is 493 m/h, as depicted in Figure 14b. In Figure 14a, a parabola can be observed, indicating that the horizontal position error improves owing to the improvement in the *z*-axis angular velocity bias. However, in Figure 14b, this trend is no longer observed because of the correction signal obtained by the gyrocompass. The bias values that are near the horizontal error of 493 m in Figure 14a are 2.153 × 10^−3^°/s (horizontal error of 491.03 m) and 2.175 × 10^−3^°/s (horizontal error of 486.20 m).

## 4. Discussion

This section briefly summarizes and discusses the results of attitude and position estimation using the proposed TG and KF methods for initial bias estimation. In Figure 9d, the initial bias for the *x*-axis by KF increases from 1 s to 220 s and then decreases to 400 s. In Figure 12b, it decreases to 139 s and then increases to 400 s, and as the subsequent fluctuations are almost the same, it can be concluded that the initial bias estimation affects the roll estimation. This trend can also be confirmed by comparing Figure 9e with Figure 12d. A comparison of Figure 9a,d reveals that for TG, the variation in the initial bias is smaller than that for KF, and the variation in the roll is also smaller than that for KF, as observed from comparing Figure 11a,b. Based on these results, we concluded that the accuracy of the initial bias estimate is one of the factors affecting the stability of the estimation accuracy of the roll and pitch and that the initial bias estimation using TG has high accuracy in the roll and pitch estimation. Comparing Figure 9c,f, it is observed that for the initial bias estimate of the *z*-axis angular velocity as well as the other axes, the variation in the TG is smaller than that in the KF. Figure 13b shows that the horizontal error of INS/DVL decreases at 377 s and 650 s because the average *z*-axis angular rate bias estimate is approximately 0.123°/s around that bias length, as shown in Figure 9f, indicating that the *z*-axis initial bias estimate affects the horizontal error when using sensors with large errors, such as MEMS sensors. In Figure 14a, the bias values that result in an average horizontal error of approximately 493 m by the IDG are 0.00215317°/s (horizontal error of 491.03 m) and 0.00217551°/s (horizontal error of 495.40 m), for a difference of 2.23 × 10^−5^°/s. Therefore, for the accuracy of this IMU, it is necessary to estimate the initial bias with an accuracy of approximately 2.00 × 10^−5^°/s or better.

For INS/DVL, the relationship between the estimated bias and the distance error is close to a quadratic curve, as illustrated in Figure 14a, and we may be able to use this trend in bias estimation. However, in Figure 14b, where the correction is taken from the gyrocompass, this trend is not observed. Therefore, in bias estimation with low-cost sensors, it is necessary to consider different estimation methods depending on whether or not the correction by the gyrocompass can be obtained. INS/DVL recorded horizontal minimum error values of 387 and 398 m, respectively, for both the TG and KF; these are both better than 466 m, which is the minimum difference in the horizontal positions for the IDG for the TG, but the possibility that the *z*-axis angular velocity bias is too large cannot be eliminated. One reason is that the accuracy of the IMU *z*-axis gyro and the accuracy of the gyrocompass make it difficult for the IMU to exceed the accuracy of the gyrocompass based on initial bias correction alone. Suppose that the initial bias of the *z*-axis gyro is set to 387 m, as shown in Figure 14a. In this situation, if the ship takes a heading opposite to the current heading, i.e., southwest, then the horizontal error could be close to 560 m (=493 m + 493 − 387 m) because the bias of the *z*-axis gyro of the INS/DVL is too large. To analyze the results, it is necessary to prepare a bow heading estimator with better accuracy than that of the gyrocompass, which is a task for future research. When the data interval of the bias estimation by the TG is 400 s or longer, the probability of less than 1852 m (1 nm: nautical mile) is 100% for INS/DVL, and the probability of less than 555.6 m (0.3 nm) is 100% for IDG. In contrast, when the data interval of bias estimation by the KF is more than 400 s, the horizontal position error is less than 1 nm with a 9% probability for INS/DVL and less than 0.3 nm with a 100% probability for IDG. However, for IDG using the TG, the variation is 2 m, whereas for the KF, the variation is 27 m. As with the angular velocity, the initial bias value for acceleration can also be more stably estimated using the TG. An RMS difference of 1.19 × 10^−2^ m/s^2^ is observed between the TG and KF for the *x*-axis acceleration. For the *y*-axis, the TG is stable at 1.15 × 10^−1^m/s^2^, whereas the KF shows greater variation. For the *z*-axis, there is an average difference of 8.26 × 10^−3^ m/s^2^. The acceleration calculated by the TG is an estimate that can satisfy the average difference of 0.04 m from the position estimated by the IGG when used in pure INS, as shown in Figure 5. Although it can be inferred that TG is superior to KF, to verify this conclusion, a highly accurate system, which is not available in our laboratory, is required to obtain the reference values of velocity in each axis direction.

## 5. Conclusions

Previously, we investigated the INS/DVL and IDG using low-cost MEMS sensors as emergency position estimation systems for situations in which the GNSS is not available. In this study, we proposed an initial bias estimation method using these low-cost MEMS sensors and evaluated the accuracy of attitude and position estimation by the INS/DVL and IDG during a 1 h navigation. It was shown that the initial bias estimate by TG can be used for more stable roll and pitch estimation than is possible with the KF. In addition, whereas the value estimated by the KF fluctuates by approximately 0.13°, that obtained using the TG is stable within 0.02°, showing almost no fluctuation. Using an estimated data interval of more than 400 s for the TG, we achieved results of 1852 m/h (1 nm/h) with a 100% probability for INS/DVL and 555.6 m/h (0.3 nm/h) with 100% probability for IDG. Because it is difficult to obtain stable position estimation accuracy with the KF, the proposed method is effective for initial bias estimation of INS/DVL and IDG using low-cost MEMS sensors. In future work, it will be necessary to improve the estimation method, including the attitude of the TG and the selection method of the data used for the initial bias. In addition, although the bow heading input by the gyrocompass is very useful, it will be necessary to consider the use of a combined GPS compass and IMU to estimate the bow heading for ships without a gyrocompass, such as ships weighing less than 500 t.

As the authors have experienced, in marine INS research, it is sometimes very difficult to know the target values because the INS equipment that can obtain the reference data is very expensive and cannot be purchased. In addition to the initial bias estimation, the proposed method can be used to obtain the estimated true values of the angular velocity and acceleration and other INS parameter calculations, even for ships without expensive INS equipment. We hope that this paper will be of assistance to those who face similar difficulties in related research.

## Figures and Tables

**Figure 1 sensors-22-05334-f001:**
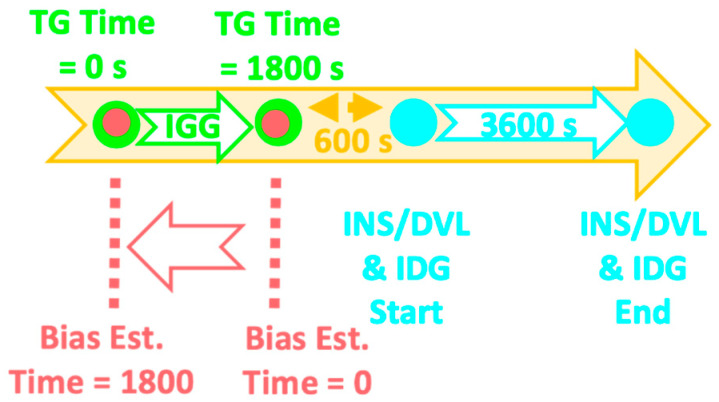
Time relationship for bias estimation.

**Figure 2 sensors-22-05334-f002:**
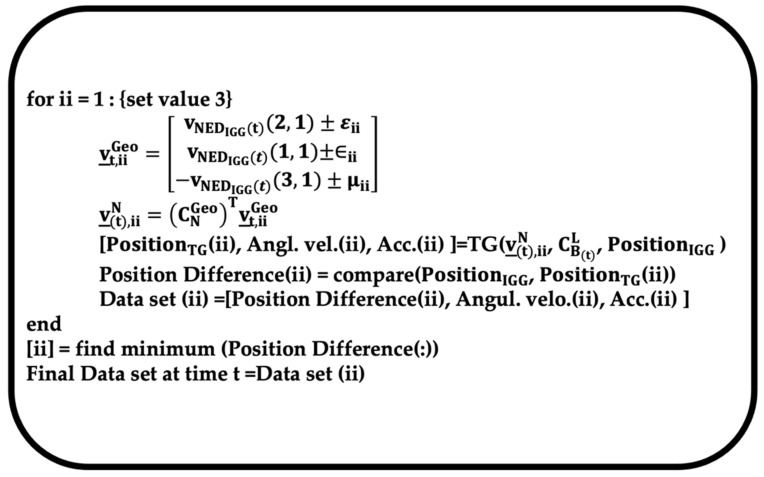
Image of processing program at a particular segment and time.

**Figure 3 sensors-22-05334-f003:**
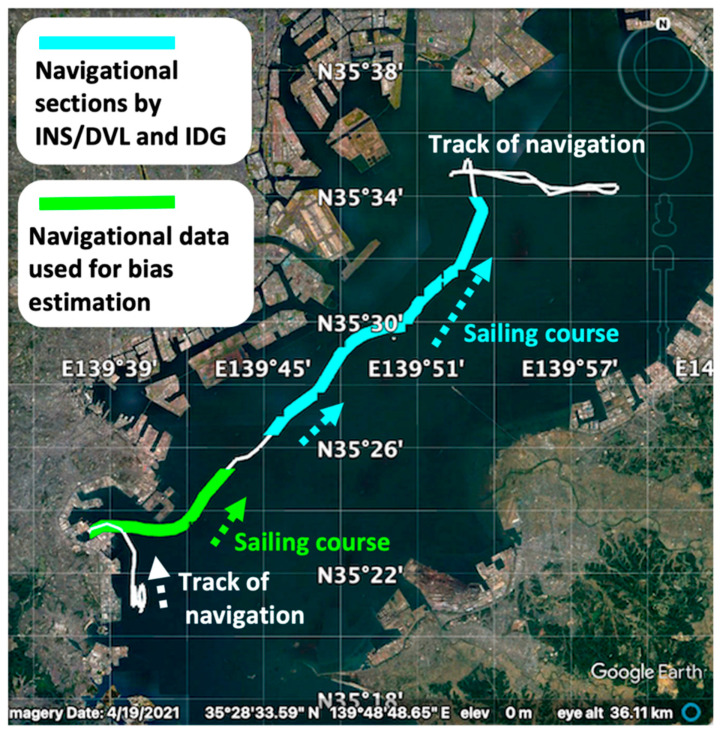
Navigation track of the experiment.

**Figure 4 sensors-22-05334-f004:**
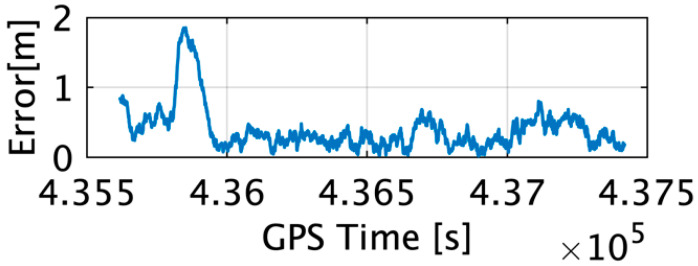
Horizontal error of the IGG estimates.

**Figure 5 sensors-22-05334-f005:**
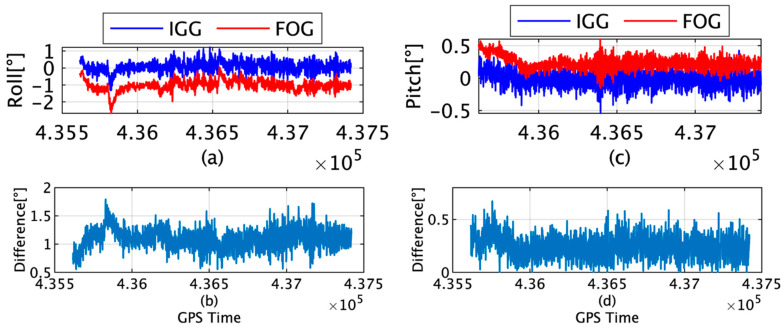
(**a**) Roll angle of IGG and FOG; (**b**) roll difference (RMS) between IGG and FOG; (**c**) pitch angle of IGG and FOG; (**d**) pitch difference (RMS) between IGG and FOG.

**Figure 6 sensors-22-05334-f006:**
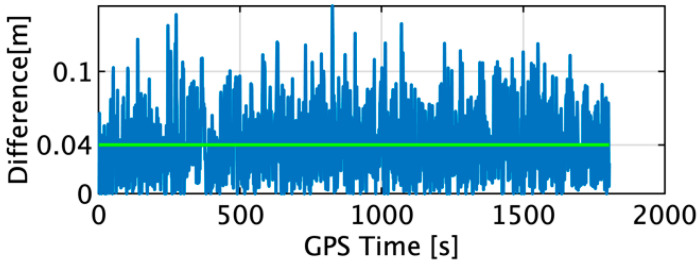
Difference in the horizontal positions estimated by the TG and IGG.

**Figure 7 sensors-22-05334-f007:**
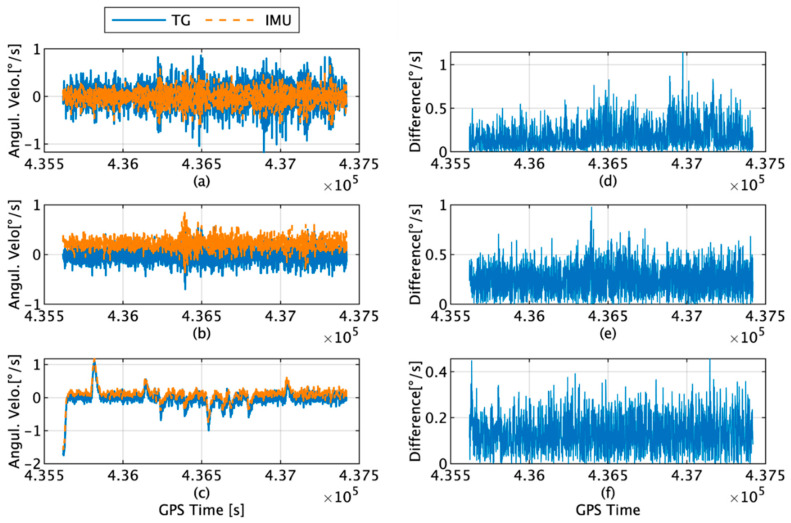
(**a**) Angular velocity along *x*-axis angular velocity of the IMU (orange) and TG (blue); (**b**) *y*-axis angular velocity of the IMU (orange) and TG (blue); (**c**) *z*-axis angular velocity of the IMU (orange) and TG (blue); (**d**) difference in the *x*-axis angular velocity between the IMU and TG; (**e**) difference in the *y*-axis angular velocity between the IMU and TG; (**f**) difference in the *z*-axis angular velocity between the IMU and TG.

**Figure 8 sensors-22-05334-f008:**
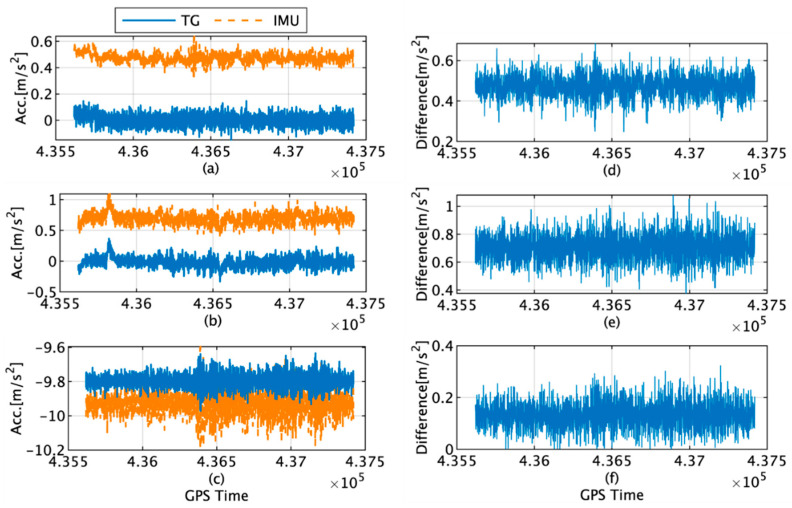
(**a**) Acceleration along *x*-axis acceleration of the IMU (orange) and TG (blue); (**b**) *y*-axis acceleration of the IMU (orange) and TG (blue); (**c**) *z*-axis acceleration of the IMU (orange) and TG (blue); (**d**) difference in *x*-axis acceleration between the IMU and TG; (**e**) difference in *y*-axis acceleration between the IMU and TG; (**f**) difference in *z*-axis angular velocity between the IMU and TG.

**Figure 9 sensors-22-05334-f009:**
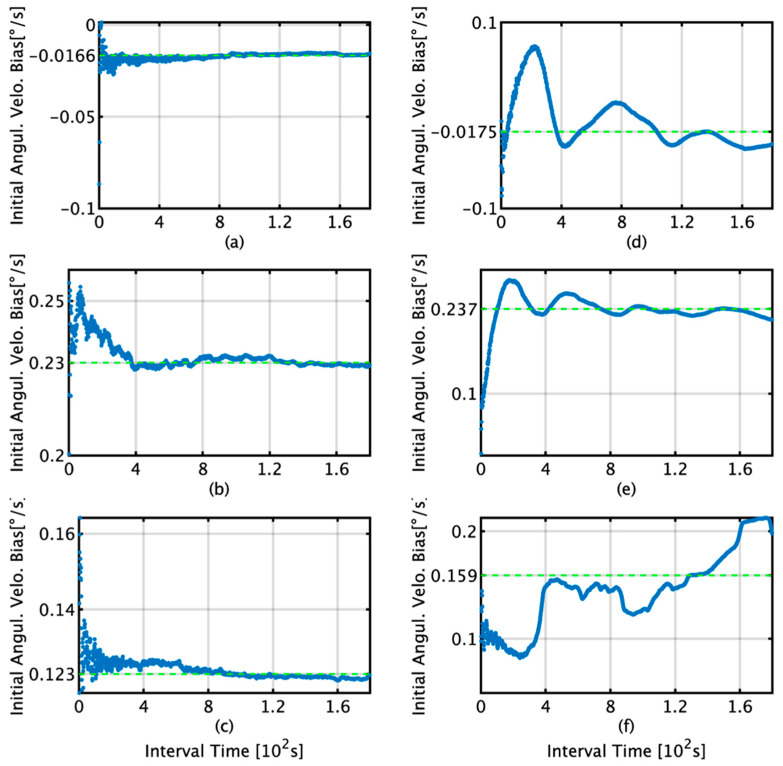
Initial bias estimates and data interval relationships obtained by the TG and KF for angular velocity; (**a**) *x*-axis angular velocity bias by TG; (**b**) *y*-axis angular velocity bias by TG; (**c**) *z*-axis angular velocity bias by TG; (**d**) *x*-axis angular velocity bias by KF; (**e**) *y*-axis angular velocity bias by KF; (**f**) *z*-axis angular velocity bias by KF.

**Figure 10 sensors-22-05334-f010:**
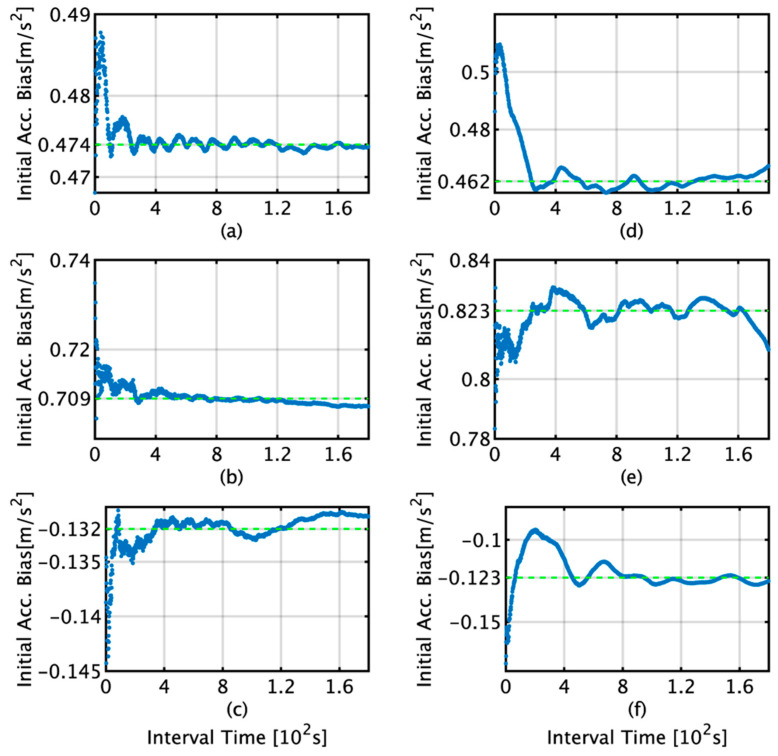
Initial bias estimates and data interval relationships obtained by the TG and KF for acceleration. (**a**) *x*-axis acceleration bias by TG; (**b**) *y*-axis acceleration bias by TG; (**c**) *z*-axis acceleration bias by TG; (**d**) *x*-axis acceleration bias by KF; (**e**) *y*-axis acceleration bias by KF; (**f**) *z*-axis acceleration bias by KF.

**Figure 11 sensors-22-05334-f011:**
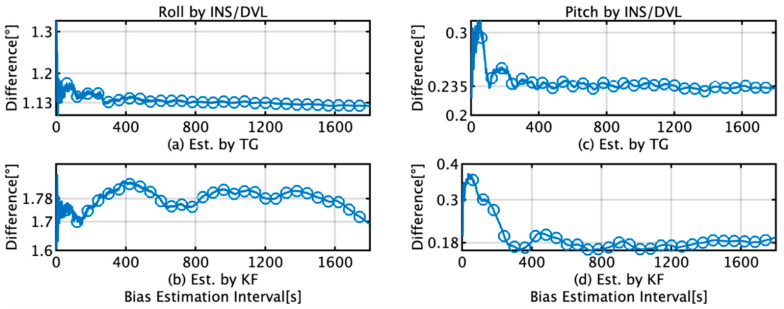
RMS difference between estimates by the INS/DVL and FOG during a 1 h voyage and the initial bias estimates from the TG and KF; (**a**) roll difference when initial bias is estimated by TG; (**b**) roll difference when initial bias is estimated by KF; (**c**) pitch difference when initial bias is estimated by TG; (**d**) pitch difference when initial bias is estimated by KF.

**Figure 12 sensors-22-05334-f012:**
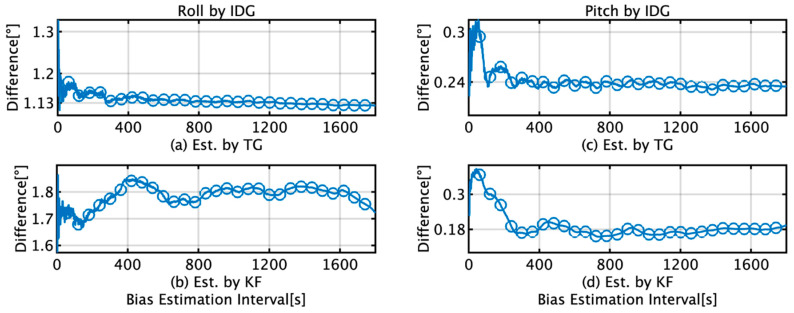
RMS difference between estimates by the IDG and FOG during a 1 h voyage and the initial bias estimates from the TG and KF. (**a**) Roll difference when initial bias is estimated by TG; (**b**) Roll difference when initial bias is estimated by KF; (**c**) Pitch difference when initial bias is estimated by TG; (**d**) Pitch difference when initial bias is estimated by KF.

**Figure 13 sensors-22-05334-f013:**
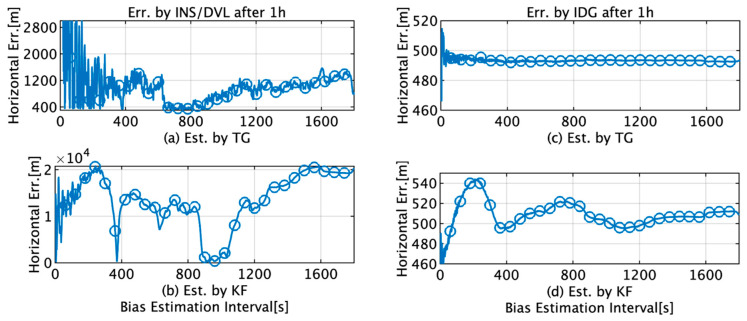
Horizontal position errors after 1 h estimated by INS/DVL and IDG with initial bias estimation by TG and KF; (**a**) horizontal error by INS/DVL with initial bias estimation by TG; (**b**) horizontal error by INS/DVL with initial bias estimation by KF; (**c**) horizontal error by IDG with initial bias estimation by TG; (**d**) horizontal error by IDG with initial bias estimation by KF.

**Figure 14 sensors-22-05334-f014:**
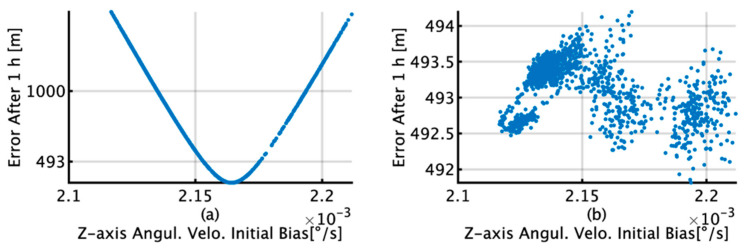
(**a**) Relationship between *z*-axis angular velocity initial bias and horizontal error after 1 h in INS/DVL; (**b**) relationship between *z*-axis angular velocity initial bias and horizontal error after 1 h in IDG.

**Table 1 sensors-22-05334-t001:** Abbreviations and full terms.

Abbreviation	Full Term
TG	Trajectory Generator
INS	Inertial Navigation System
DVL	Doppler Velocity Log
IDG	INS/DVL/Gyrocompass
IGG	INS/GPS/Gyrocompass
KF	Kalman Filter

**Table 2 sensors-22-05334-t002:** Sensor information.

	GNSS	IMU			DVL	FOG
Name	Trimble SPS855	CSM-MG100			ATLAS DOLOG SYSTEM	JCS7402-A
Freq.	5 Hz	100 Hz			1 Hz	1 Hz
Accuracy	Position	Gyro	Acc.	Position	Speed	Roll and Pitch
	<0.1 [m]	±0.00175 [rad/s]	±0.01 [m/s^2^]	3.0 m RMS or Less	0.01 [knot] or 0.2% of the measured value	≤±0.15° at input ≤±10° ≤±(0.2° + 1% of input) at input = ±10°–45°

**Table 3 sensors-22-05334-t003:** Specifications of the TG-5000 gyrocompass.

Setting Time	Within 2 h	Accuracy on Scorsby Table	≤±0.5°
Setting Point Error	≤±0.3°	Repeatability of Setting Point	≤±0.2°
RMS Value	≤0.1°	Accuracy Under Environmental Variation	≤±0.5°

**Table 4 sensors-22-05334-t004:** Maximum and minimum differences in the RMS roll and pitch when using the initial bias estimates from the TG and KF for INS/DVL.

	Roll by ISN/DVL (°)					Pitch by INS/DVL (°)				
	Total		Over 400 s			Total		Over 400 s		
	Max.	Min.	Max.	Min.	Ave.	Max.	Min.	Max.	Min.	Ave.
By TG	1.33	1.10	1.15	1.12	1.13	0.31	0.22	0.24	0.23	0.24
By KF	1.86	1.60	1.83	1.69	1.78	0.37	0.16	0.20	0.16	0.18

**Table 5 sensors-22-05334-t005:** Maximum and minimum differences in the RMS roll and pitch when using the initial bias estimates from the TG and KF for IDG.

	Roll by IDG (Degree)					Pitch by IDG (Degree)				
	Total		Over 400 s			Total		Over 400 s		
	Max.	Min.	Max.	Min.	Ave.	Max.	Min.	Max.	Min.	Ave.
By TG	1.33	1.11	1.14	1.12	1.13	0.31	0.23	0.24	0.23	0.23
By KF	1.86	1.57	1.85	1.72	1.80	0.39	0.15	0.20	0.15	0.18

**Table 6 sensors-22-05334-t006:** Maximum and minimum differences in the horizontal positions for the INS/DVL for the TG and KF.

	Err. by INS/DVL after 1 h (m)					Err. by IDG after 1 h (m)				
	Total		Over 400 s			Total		Over 400 s		
	Max.	Min.	Max.	Min.	Ave.	Max.	Min.	Max.	Min.	Ave.
By TG	1.76 × 10^4^	340	1569	340	921	515	466	494	492	493
By KF	2.08 × 10^4^	217	2.06 × 10^4^	345	1.31 × 10^4^	544	442	522.00	495	507

## Data Availability

The data presented in this study are available on request from the corresponding author. The data are not publicly available due to the matter of the training ship. When a request is received for a reasonable reason, it can only be provided after explaining it to the relevant department and obtaining permission from all departments.

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
