# Peer review of "Application of Initial Bias Estimation Method for Inertial Navigation System (INS)/Doppler Velocity Log (DVL) and INS/DVL/Gyrocompass Using Micro-Electro-Mechanical System Sensors"

_sensors, 2022, doi:10.3390/s22145334_

Round 1
Reviewer 1 Report
The authors proposed an initial bias estimation method that can be used for vessels. The key of the method is to use TG to solve the angular velocity, acceleration and gyrocompass sensor to suppress the drift of IMU. And the authors carried out relevant experiments to verify them. Here are some comments:
1. The description of formula (6) is unclear.
2. What does IGG mean? Many related formulas are not clearly stated, such as line 128: the explanation of formula symbols is vague.
3. The derivation description of the formula is not systematic. Although it is concise, it is very confusing. For example, in formula (9), H matrix suddenly appears, although the author cites [8].
4. In the Section of 2, the author should provide a pipeline for the overall estimation of bias algorithm, as well as the hardware configuration diagram, coordinate system and other frameworks. Otherwise, it would be chaotic for readers.
5. It seems common to use gyrocompass to correct heading angle drift. The author should explain the advantages and algorithms used. The description in Section 2.3 is not clear.
6. How is formula (12) derived?
7. Line 174-175, why does the use of IGG values have the effect of smoothing position and posture? How is smooth?
8. The essence of formulas (13) to (24) is to use the Euler angle to turn to the direction cosine matrix. This is not the focus of the research. It is not necessary to spend so much space showing the base mathematical formula.
9. Line 247: “From the above, we believe that real-time processing is also possible, but in this study, we present the results as post-processing analysis results.”. How can you prove that you believe?
10. The ship is running on a horizontal plane, the roll and pitch angle is ideally zero. Take roll as an example: Figure 4 shows that the range of ordinates is , but the maximum differences is . This is not a strong demonstration of accuracy, and the verification is not sufficient. What about the accuracy if the roll and pitch become larger?
Is there any simulation or other quantitative experiment to show the accuracy? It is suggested that the Figure 5 of line 282 be similarly demonstrated or modified.
11. Line 275: Figure5 Difference in the Horizontal Positions Estimated by the TG and IGG. Is there a global trajectory comparison diagram?
12. Line 322-343, 347-368, etc. For the comparison of indicators, it will be more concise and intuitive to use tables and other methods to describe, and the text expression is redundant.
13. line 321: The title of 3.3.1 is not very ideographic.
14. Line 400-409: For INS/DVL, the minimum error was 340 m/h, as shown in Figure 14 (a), and for IDG, the minimum error was 492 m/h and the average error was 493 m/h, as shown in Figure 14 (b). Why does the error become larger when the gyrocompass is added to suppress the angular velocity drift? Line438 - 453 has a weak explanation for this.
15. Line 433-437: Figure 14 (a) indicates INS/DVL. Why is IDG on line 434? What is the basis for obtaining accuracy from two bias value?
16. Line 147: the and the gyrocompass correction is given by. What does it mean?
17. Some typos/errors:
line 162: equantion whih.
Section 2.2 is written as Section 2.3.
Line 225 S or A, inconsistent with Table 1.
Figure 5 and figure 13 have more than one.
Line 289 and 310: units not suitable.
Line 346 with or without?
18. Punctuation:
Line 167: INS/DVL/.
Formula symbol uniformity, such as formula (24), GEO or Geo.
19. Based on the comments about typos and punctuation problems, the paper still has some similar problems. It is strongly recommended that the author should read the article carefully again. In addition, the English expression ability also needs to be strengthened.
Author Response
Thank you very much for your important concerns. I reply to you according to your questions. Revised parts in the manuscript were highlighted in green in the attached manuscript. Please note that English editing was conducted by professional editors at commercial company.
Please see the attachment.

Reviewer 2 Report
About the literature and format:
1. The authour should pay attention to the format and the position of the graphics and tables. The typesetting of the paper is somehow disorder.
2. There are some mistakes in words and phrases, for example in line 162, the word "which" is apparently wrong typed in "whih" and the word "equation" is wrongly typed in "equantion". Similar situation also appear in line 167, the abbreviation seems not complete. The author must carefully check these kinds of careless mistakes.
3. For friendly reading, it is a good idea for author to add an abbreviation reference table.
About the scientific persuasiveness:
1. What is difference between IDG and IGG? I found the explanation INS/DVL/Gyrocompass (IDG), but I did not find the meaning of IGG. Does IGG mean INS/GPS/Gyrocompass? If so please add explanation or an abbreviation reference.
2. The length of each section is not allocated properly. The author pay too much attention on the experiments analysis but little works on mathmatic derivation.
3. No offence, I have to say the innovation and novelty of the proposed method are not very high. The mathmatic model of inertial calculation and the trajectory-based inverse inertial calculation methods have already been all well-researched. The author's work is using the well-researched trajectory-based inverse inertial calculation methods to estimate the initial bias of the IMU. Typically, there are two main method to obtain the bias: pre-calibration and online calibration. The author may read these two articles below, which concretely introduce some calibration method pre or real time:
(1) S. Bhatia, H. Yang, R. Zhang, F. Höflinger and L. Reindl, "Development of an analytical method for IMU calibration," 2016 13th International Multi-Conference on Systems, Signals & Devices (SSD), 2016, pp. 131-135, doi: 10.1109/SSD.2016.7473706.
(2) G. G. Scandaroli and P. Morin, "Nonlinear filter design for pose and IMU bias estimation," 2011 IEEE International Conference on Robotics and Automation, 2011, pp. 4524-4530, doi: 10.1109/ICRA.2011.5979795.
Pre-calibration face the problem of the repeatable stability, online calibration face the problem of introduce the error of other sensors. The proposed method seems neither real-time nor avoiding introducing the error of other sensors. Although the author claims that the method has a better performance than classic KF method, I think it is a nice trial but is not useful or has practical meaning for real utility. The proposed algorithm is just a recurssive matching algorithm and I don't find any novelty. To say the least, even the matching algorithm is some how innovative for the its new application, the author should give the criterion or cost function to judge the least difference point. I think it is one of the key points of the proposed algorithm. Is it the Euclid Distance or other criterion? If the author want to do some further research on online calibration, my advice for him/her is focusing on "real-time" or "less error sources". Maybe for maritime application, the ZUPT method is a better choice for your application.
4. Can the author explain, why the RMSE of pure IDG calculation is not in a accumulated pattern. Generally, the navigation results of INS/DVL method always perform with accumulating error.
Author Response

(The authors gave the same response as above.)

Reviewer 3 Report
This paper studies the application of INS/DVL and INS/DVL/Gyrocompass based on MEMS inertial sensors. The author tries to improve the performance of these applications by estimate the initial bias error. The article has done some work on the experiment, but the innovative description of the article is not prominent, and the writing is not standardized. Before this article reaches the quality that can be published, there are still issues that must be improved.
1. The expression of the article must be greatly improved, and the label of Section 2.2 is omitted. The position and size of the pictures in the paper should be consistent. Two center alignment methods are used in the manuscript. And the labels of the pictures are also wrong. The author should at least confirm these basic issues before submitting the manuscript.
2. In the introduction section, I did not find any description of the contribution made by this paper. The author should clearly explain the problem this paper aims at and what contributions have been made.
3 The article use the Section 2 to describe the specific research content and methods, but in this section, which part is the existing research method, and which part is the author's contribution have made?
4 The content of Conclusions is a summary of the paper, not the motivation for researching this issue, and author should reorganize this section.
After the above problems are solved, the quality of the article can be further evaluated.
Author Response

(The authors gave the same response as above.)

Round 2
Reviewer 2 Report
There is no further suggestion to the author.
Author Response
We have not received any comments from you at the 2nd peer review, however, we recognized that all check items were marked as "Can be improved" and not sufficiently corrected.
We have further revised the paper based on the 1st round peer-review results and the points raised by the other reviewers. Revised parts in the manuscript were highlighted in blue in the attached manuscript.We wish to express our appreciation to the reviewers for their insightful comments on our paper. The comments have helped us significantly improve the paper.
Reviewer 3 Report
In general, the presentation of the paper has been improved after a comprehensive revision, but there are still some apprarent problems.
1) There are remains some problems with the numbering and typesetting of the formulas, such as the alignment between (3) and (4), and dose the formulas between (8)-(9) have numbers? why (10) is followed by (5)-(6)? And so on.
2)In lines 128-142, the author cost a lot of space for the explanation of the formula (1)-(4). Although they are not difficult formulas, the explanation is very long and difficult to read.
3)Authors should clearly state their contributions in the manuscript at the end of the introduction, preferably in several points, if possible.
4) In the second chapter, there is still a problem of presentation. It can be seen that parts 2.1 and 2.3 are the initial bias estimation scheme proposed by the author, but the description of these two parts is too fragmented, author spends most of the space listing the related formulas of the error equation that have been derived from other papers, and this part of the content is not the main focus of the author. I think the schematic diagram that probably best describes the author's scheme is the Figure 8 in the experimental chapter?
In general, the author has made some efforts to improve the application performance of integrated navigation such as MEMS-INS/DVL, but the author's expression and article arrangement should be the most obvious shortcomings.
Author Response
Thank you very much for your important concerns. I reply to you according to your questions. Revised parts in the manuscript were highlighted in blue in the attached manuscript. Please note that English editing was conducted by professional editors at commercial company. Please see the attachment.
